# Experiences that influence how trained providers support women with breastfeeding: A systematic review of qualitative evidence

**Mary Jo Chesnel** [ORCID]**\*, Maria Healy, Jenny McNeill**

School of Nursing and Midwifery, Medical Biology Centre, Queen's University Belfast, Belfast, Northern Ireland

\* mchesnel01@qub.ac.uk

**Data Availability Statement:** All relevant data are within the manuscript and its Supporting Information files.

## Abstract

### Introduction

There is a need to improve breastfeeding support interventions as although many are evidence-based, a sequential increase in breastfeeding rates is not evident. It is crucial to understand why the implementation of evidence-based guidelines in practice does not always translate to positive experiences for women and improve breastfeeding rates. This systematic review aims to synthesise breastfeeding support experiences of trained support providers and their impact on breastfeeding support practices.

### Methods

A strategy was developed to search seven databases including Medline and CINAHL and grey literature for qualitative studies. Studies eligible for inclusion reported professional and trained peer experiences of supporting women to breastfeed. PRISMA guidelines were followed and included studies were quality appraised using the CASP Qualitative Checklist. A thematic synthesis of included studies was undertaken and confidence in the review findings was assessed using the CERQual tool. The study protocol, registered in the International Prospective Register of Systematic Reviews PROSPERO registration number: CRD42020207380, has been peer reviewed and published.

### Findings

A total of 977 records were screened, which identified 18 studies (21 papers) eligible for inclusion comprising 368 participants. Following quality appraisal, all studies were deemed suitable for inclusion. The thematic synthesis resulted in four analytical themes: 1) A personal philosophy of breastfeeding support 2) Teamwork and tensions in practice 3) Negotiating organisational constraints and 4) Encounters with breastfeeding women. Findings demonstrated that a range of experiences influence practice, and practice evolves on continued exposure to such experiences. The potential of each experience to facilitate or inhibit breastfeeding support provision is fluid and context specific.

**Funding:** The authors received no specific funding for this work.

**Competing interests:** The authors have declared that no competing interests exist.

## Conclusions

Experiences, as named above, are modifiable factors contributing to the development of a philosophy of breastfeeding support based on what the provider believes works and is valuable in practice. Further research is required into the range of factors which underpin context-specific breastfeeding support practice, to improve both women's experiences and intervention effectiveness.

## Introduction

Global breastfeeding rates fail to reach the World Health Organization (WHO) target of 50% exclusive breastfeeding until 6 months of age by 2025 [1]. This occurs despite a wealth of evidence of the health risks to mother-infant dyads of not breastfeeding [2–4], and recognition that breastfeeding is key to meeting the United Nations Sustainable Development Goals 2 and 3 for 2030 aiming to end hunger, improve nutrition and promote health [5]. Low rates of duration and exclusivity of breastfeeding are more prevalent in high-income countries (as classified by World Bank ratings) despite a range of public health interventions focusing on promoting and protecting breastfeeding [3, 6–8]. Further research into breastfeeding support interventions is required in order that the intended outcomes of increasing breastfeeding rates and positive experiences for women are more easily achieved [1, 9].

Evidence from existing reviews demonstrate that organised support from trained breastfeeding supporters, whether lay or professional, can prevent early unintended breastfeeding cessation [3, 10, 11]. Breastfeeding support is regarded as a complex intervention which involves sharing of advice and information, provision of skilled help, reassurance and increasing the mother's confidence [12] within interpersonal interactions. Healthcare staff involved in maternal and child health services, lactation consultants and trained volunteers who have received training on how to support women to breastfeed are key to implementing effective breastfeeding support services.

Supporting the women who want to breastfeed to meet their breastfeeding goals as part of an evidence-based intervention appears to be a simple task. However, many women cease breastfeeding before they intended to do so [13–15] and some women are disappointed with the support they receive from trained providers [16–18]. Little attention is paid to the influence of factors other than evidence-based research on healthcare professional practice in relation to breastfeeding [19]. It is important to explore this knowledge gap in the context of all trained breastfeeding support providers, whether lay or professional. Indeed, the Medical Research Complex interventions framework [20] highlights the importance of the context of intervention implementation, as much as the intervention itself for successful outcomes.

Several systematic reviews have examined breastfeeding support interventions, but these studies prioritised effectiveness outcomes in terms of breastfeeding rates [3, 6, 21–23]. Such evidence is valuable but does not closely examine the underlying context of supporter behaviour. To improve provision of breastfeeding support, questions now need to be asked about contextual factors that may influence the practices of the support providers.

Relevant qualitative reviews in this area have found that, although breastfeeding support is viewed as important by both women and midwives, it can be difficult to implement the intervention effectively and compassionately. Schmied *et al.* [2011] demonstrated that women experience breastfeeding support along a continuum from a woman-centred 'authentic presence' to 'disconnected encounters' which reflect a lack of relational-based support from

providers [16]. Swerts *et al.* [24] explored midwives' perceptions of their role in support provision, reporting variance in how midwives support women with their breastfeeding [24]. These authors propose that while midwives mostly practice as a 'technical expert' they would prefer to practice as a 'skilled companion', however this requires an appropriately supportive working environment. Gaps in current knowledge persist as little is known about the factors that influence practice of the range of both healthcare professionals and trained lay breastfeeding supporters who provide routine, everyday breastfeeding support. Exploring experiences which may influence routine practice across a range of breastfeeding support provider roles is novel in relation to population and outcome, while building upon previous studies to advance the work of improving breastfeeding support provision for women. This systematic review sought to answer three research questions:

1. What is known about experiences that influence provision of breastfeeding support?

2. How do the experiences of trained breastfeeding support providers influence their breastfeeding support practices?

3. Which experiences facilitate or impede provision of breastfeeding support to women?

## Methods

PRISMA guidelines were followed in conducting this review [25]. The review protocol is registered in the International Prospective Register of Systematic Reviews PROSPERO registration number CRD42020207380 and has been published [26].

### Definitions

**Trained breastfeeding supporters.** Trained breastfeeding support providers refers to trained healthcare staff working with breastfeeding women and healthy infants/children as part of their role, and non-healthcare breastfeeding support providers such as peer breastfeeding supporters and lactation consultants who have undertaken formal accredited training. Students, untrained volunteers, healthcare staff working with sick infants/children were not included in the review as the focus was on routine breastfeeding support for mothers with healthy babies.

### Breastfeeding support

The term "breastfeeding support" in this review refers to proactive or reactive interactions between women, infants and trained breastfeeding support providers offering reassurance, praise, skilled help, problem solving, information and social support in face-to-face, group or digital settings such as social media groups, telephone calls or text messages. Support may be provided in acute hospital, maternity units, primary care, voluntary and community settings and women's own homes. This definition is adapted from McFadden *et al.* (2017). This systematic review focuses on routine breastfeeding support in the absence of complication.

### Search strategy

A systematic search strategy was developed in collaboration with an expert subject librarian guided by a PEOT [27] question format. For the purposes of this review, Context replaces Outcome in the mnemonic as outcomes are not directly measurable in qualitative studies: Population (trained breastfeeding support providers as per study definition above), Exposure

(breastfeeding and breastfeeding support provision), Context (experiences that influence breastfeeding support practices), Type of study (studies that have qualitative methods or findings).

## Study selection

### Inclusion criteria

Qualitative studies and mixed methods studies with qualitative methods and findings were included. Study findings were required to focus on trained breastfeeding support providers' personal and professional experiences (emotions, past encounters, training, practice) in relation to breastfeeding and supporting women to breastfeed, and the influence of those experiences in providing breastfeeding support.

Reporting of ethical committee approval and evidence of data to support findings was required. The population comprised of trained breastfeeding support providers who provide routine breastfeeding support to healthy women with healthy infants in high income countries as defined by the World Bank [28]. Studies of the experiences of supporting women with breastfeeding in acute hospital, maternity units, primary care, voluntary and community settings and women's own homes were included.

### Exclusion criteria

Mixed-methods studies that did not report qualitative findings were excluded. Studies of students, untrained volunteers and healthcare staff working with sick infants/children were not included as the focus of this review is on routine breastfeeding support for healthy mothers with healthy infants. Breastfeeding support was not considered routine when delivered to women with additional care needs [29] or delivered in a neonatal or paediatric setting. Studies with heterogeneous samples including, for example, school nurses or paediatricians were excluded if data pertaining to the experiences of trained breastfeeding support providers routinely working with women and healthy infants/children could not be isolated from dataset. Studies from low-income countries were excluded.

The search was undertaken using CINAHL +, MEDLINE ALL, Maternity and Infant Care, EMBASE, APA PsycINFO, Web of Science and Scopus databases. Reference lists of retrieved eligible studies were hand-searched for further eligible studies. An English language restriction and a methodological filter for qualitative studies was included. The reference lists of unpublished literature sourced via Open Grey and British Library Ethos were searched for relevant published studies. The search period included year 2003 –current, with the latest search completed on 16th June 2021. The start year of 2003 was chosen in order to identify research undertaken following publication of the World Health Organization's Global Strategy for Infant and Young Child Feeding (2003) [30] which advised that women exclusively breastfeed for 6 months and continue breastfeeding for two years and beyond for optimal health benefits to mother and infant. A search strategy based upon MeSH headings, related keywords and truncations was developed for each database. Boolean Operators OR and AND were used with the search terms. The search strategy is given in S1 Table of Search histories.

### Study selection procedure

Studies were selected for inclusion following a two-stage process using *Covidence* software. Findings from the searches were exported to *Covidence* via EndNote X9 reference management system enabling de-duplication of records and teamwork amongst the three reviewers. Firstly, title and abstracts were screened by the first author MJC with verification by another

independent reviewer (JM or MH). Secondly, all three reviewers screened full texts independently (MJC, JM, MH).

## Data extraction

A data extraction form adapted from Healy *et al*. [31] was developed to capture information on each study's key characteristics including study location, aim, participant demographics, methodology and method, and main findings. All text in the Findings sections of the papers, alongside verbatim quotes elsewhere in the papers, were extracted if relevant to the three research questions of the review. Meaningful sections of text were extracted that identified experiences (emotions, past encounters, training, practice) that influenced how breastfeeding support was practiced by the provider, from their personal perspective, or as observed by researchers in the included study using a discourse analysis method [40]. Data extraction was carried out by MJC and reviewed by JM & MH.

## Reflexive note

The review team are all midwives and mothers who have supported women to breastfeed as part of their career. All believe that effective breastfeeding support can enable breastfeeding which is important for the physical and psychological well-being of the mother-baby dyad. Discussion within the team was used to minimise any bias due to undue focus on study findings that aligned with personal views.

## Quality assessment

All studies were critically appraised by MJC with discussion amongst the review team. The Critical Appraisal Skills Programme (CASP) Qualitative checklist tool [32] was used to assess the quality of the studies. The COREQ tool [33] was used to assess the comprehensiveness of the reporting of study design, analysis and findings. The CERQual tool [34] was used to assess confidence in the review findings as an overall body of knowledge.

# Analysis

A systematic three-step approach to analysis was used to develop themes relevant to understanding which experiences of breastfeeding and breastfeeding support influence practice. A thematic synthesis as described by Thomas and Harden [35] was conducted. An inductive approach was chosen as the authors had no assumptions about what the dataset would reveal. Thematic synthesis facilitates an interpretation of concepts across different types of intervention, which is appropriate for breastfeeding support research because the intervention is carried out in multiple formats and settings. Thomas and Harden propose that thematic synthesis is the process of recognising cross-cutting concepts across studies, even though they may not be expressed using identical words, in order to provide new insights into policy, practice and further research [35]. Principles of thematic analysis [36–38] were used throughout coding and theme development.

Coding was conducted primarily by MJC and reviewed by the co-authors. Firstly, line-by-line coding of all relevant text extracted from the Findings sections of the papers was undertaken. Excel software was used to manage data. Resultant free codes were transcribed together with supportive verbatim text to enable ease of searching and linking to data in the studies. Descriptive themes were developed in an iterative process, moving forwards and back between suggested commonalities and disparities in the meanings of the free codes. Potential descriptive themes were discussed amongst the review team and compared across studies. Analysis of

the descriptive themes led to development of four core analytical themes which reflected the synthesis of all papers in the review, agreed by all members of the review team.

## Findings

### Included studies

The database search resulted in 1811 records, of which 834 duplicates were removed leaving 977 records for screening by title and abstract. Exclusion of 933 records following screening left 46 records for assessment, including two additional records retrieved from reference list sources. From these 46 records, 25 were excluded for reasons of wrong population (e.g. not providing routine breastfeeding support), wrong exposure (no data on breastfeeding or breastfeeding support), wrong context (no data on experiences that influence breastfeeding support practices) or wrong type of study (pilot evaluations) and 21 records were included in the review. A flow diagram adapted from the Preferred Reporting Items for Systematic Reviews and Meta-Analysis (PRISMA) guidance [25] was used to report the study selection process. Fig 1 shows the study selection process.

### Study characteristics

The dataset included 300 trained healthcare staff with identified roles: 210 midwives, 16 Health Visitors, 14 Public Health Nurses, 12 Maternal Newborn Nurses and 9 Post-partum nurses. Three studies did not specify individual role prevalence in sample groups, with one describing the sample as 20 primary healthcare professional participants comprising nurses, midwives and family physicians [39], another sample was described as 10 ward staff who were trained to support women to breastfeed including midwives, maternity care assistants and nursery nurses

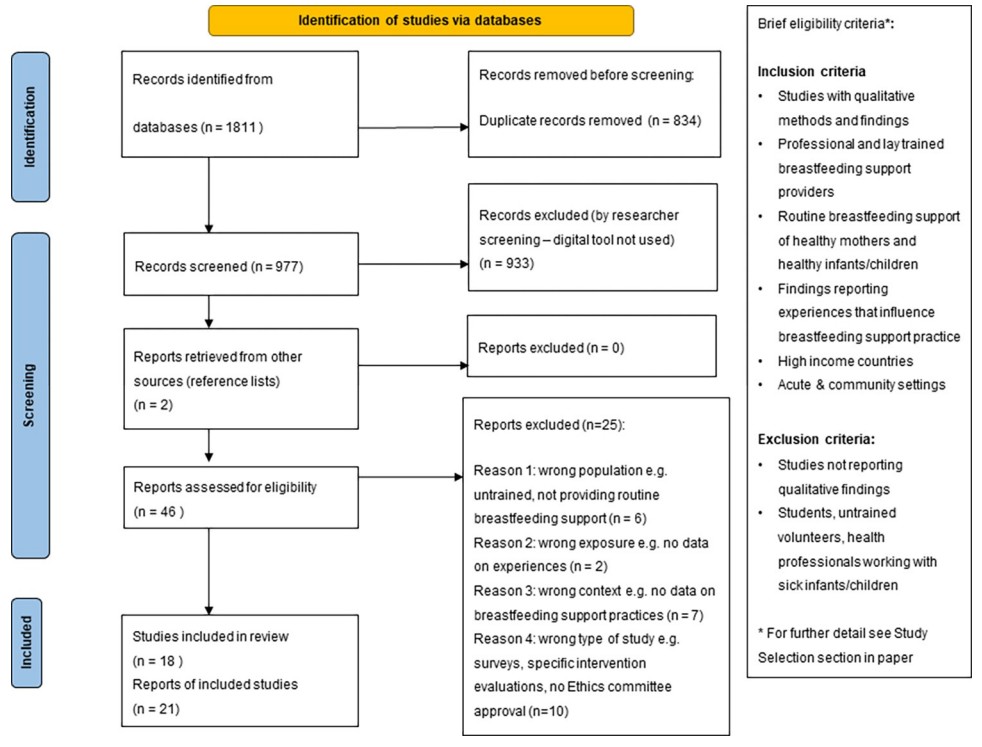

**Fig 1. PRISMA flow chart.**

working on maternity ward [40] and sample group of 9 doctors and nurses was reported in a third study [41]. The trained breastfeeding support providers in non-healthcare roles in the data set comprised 56 lactation consultants and 12 peer supporters.

Eligible studies were published between 2005 and 2020 and conducted in 8 high income countries. Included papers reported qualitative designs with either interviews or focus groups apart from one Australian solely observational study [42]. Several studies used participant observation alongside interview and focus group methods. Research took place within hospitals, community primary care, community and voluntary settings and private practice. Two studies reported UNICEF Baby Friendly accreditation [43] of their setting [44, 45]. Furber's UK based studies [46–48] reported that their setting was not Baby Friendly accredited, and all other included studies did not report Baby Friendly Accreditation status.

Data was extracted from each study comprising Title, Author, Year, Country, Study aim or research question, Practice setting, Methodology and method, Population and sample size, COREQ score and Findings (Themes/subthemes). Study characteristics and quality appraisal scores are summarized in Table 1.

## Critical appraisal

The Critical Appraisal Skills Programme (CASP) Qualitative checklist [32] was used to initially assess the quality of each study. A numerical score for quality was not assigned, rather, the tool prompted focused reading of papers regarding potential methodological limitations. The quality of reporting in the studies ranged from a score of 16 to 27 out of a possible 32 items in the COREQ checklist. Most studies did not report on researcher reflexivity. Description of analytical methods was limited in five of the studies. Information from the COREQ and CASP assessments enabled trustworthiness in the findings of each study to be gauged, and informed subsequent assessment of confidence in the findings using the Confidence in the Evidence from Reviews of Qualitative Research tool (CERQual) [49–53]. Overall, the strengths of the studies lay in the congruence of research aims and objectives with the study design.

## Themes

This review aimed to answer three research questions, asking what is known about experiences that influence provision of breastfeeding support, how such experiences influence breastfeeding support practices, and which experiences facilitate or impede provision of breastfeeding support to women. The findings of this review suggest that a range of prior and current experiences relating to breastfeeding and breastfeeding support provision, reported here as analytical themes, are very likely to contribute to breastfeeding support practices, either facilitating evidence-based compassionate care or hindering such provision. Synthesis of the data resulted in the generation of 85 free codes, eleven descriptive themes and four analytical themes supported by summary statements (Table 2). The prevalence of the descriptive themes and analytical themes in the papers is reported in S2 Table. Confidence in the review findings was assessed using the CERQual tool [34]. There was high confidence in three of the descriptive themes. Confidence was downgraded to moderate (five descriptive themes) or low (three descriptive themes) when there was concern about any of the four components of the CERQual assessment [34] methodological limitations, coherence, adequacy of data and relevance as reported in S3 Table. The analytical themes are titled *1) A personal philosophy of breastfeeding support 2) Teamwork and tensions in practice 3) Negotiating organisational constraints* and *4) Encounters with breastfeeding women.*

For brevity, exemplar quotes of data are presented below and further supporting quotes for each descriptive theme are found in the CERQual Evidence table (S3 Table).

## A personal philosophy of breastfeeding support

A personal philosophy of breastfeeding support was the most dominant analytical theme of the review. It is comprised of three descriptive themes, further detailed in subsequent sections, which show that breastfeeding support is delivered according to the providers prior experiences, established beliefs and preferences in applying knowledge for practice from various sources. Participants in all studies spoke of experiences which were developed into 3 interlinked descriptive themes to form a personal philosophy of breastfeeding support: *Personal breastfeeding experience* [39, 40, 42, 44, 45, 54–60] rated by CERQual assessment as having a high level of confidence in the finding, *Belief in the value and process of breastfeeding* [39–42,

**Table 1. Study characteristics and quality rating.**

| Title, Author & Year | Country | Study aim/ Research question | Practice setting | Methodology & Method | Population & Sample size | COREQ score | Findings: (Themes / sub-themes) |
|---|---|---|---|---|---|---|---|
| **Lactation Consultant's Perceived Barriers to Providing Professional Breastfeeding Support.** Anstey *et al.* (2018) | U.S.A. | To explore the perspectives of lactation consultants about early (prior to 4 weeks postpartum) breastfeeding problems that may lead to early weaning and identify factors that hinder the professional management of these problems. | A range of practice settings including hospitals, WIC Clinics (a nutrition program for women infants and children), private practice and pediatric offices. | Grounded theory Interviews | International Board Certified Lactation Consultants (IBCLCs) n = 30 | 27 | Two categories of factors (direct and indirect) act as facilitators or barriers to IBCLC role enactment: **Indirect barriers** (social norms, knowledge, attitudes) **Direct occupational barriers** (institutional constraints, lack of co-ordination, poor service delivery) **Direct individual barriers** (mother's social support, mother's self-efficacy) |
| **Two sides of breastfeeding support: experiences of women and midwives.** Bäckström *et al.* (2010) | Sweden | To investigate women's experiences and reflections of receiving breastfeeding support, and midwives' experiences and reflections of giving breastfeeding support. | Hospital maternity unit | Qualitative inquiry using content analysis Interviews | Midwives n = 4 | 17 | Individualised breastfeeding support increases confidence and satisfaction: **The unique woman** (Confirmation as a person and as a breastfeeding woman, Support to women whether breastfeeding or not.) **The sensitive confirming process** (observation and confirmation, practical and physical support) **Consistency of ongoing support** (Establish continuity, follow up) |
| **Liquid gold from the milk bar: Constructions of breastmilk and breastfeeding women in the language and practices of midwives.** Burns *et al.* (2012) | Australia | To examine how midwives represent breastmilk and construct the breastfeeding woman in their interactions with women during pregnancy and following birth. | Two hospital sites including hospital-based standard care, home-base post-natal care and continuity of care models | Qualitative Inquiry using discourse analysis Observation and interviews | Midwives n = 76 | 16 | **Colostrum: an atomic bomb of nutrients Mature breastmilk as superior 'liquid gold' Breastfeeding women as 'operators' of breastfeeding equipment** |

*(Continued)*

**Table 1.** (Continued)

| Title, Author & Year | Country | Study aim/ Research question | Practice setting | Methodology & Method | Population & Sample size | COREQ score | Findings: (Themes / sub-themes) |
|---|---|---|---|---|---|---|---|
| **Mining for liquid gold: midwifery language and practices associated with early breastfeeding support.** Burns *et al.* (2013) | Australia | To examine the nature and impact of the language and practices of midwives when providing breastfeeding support to women in the early post-partum period. | Two hospital sites including hospital-based standard care, home-base post-natal care and continuity of care models | Qualitative inquiry using discourse analysis Observed interactions, Focus groups and interviews | Midwives n = 76 as per Burns *et al.* (2012) | 22 | **Mining for liquid gold** (Hands on–rights to the 'equipment', The tools of the trade, facilitating and fixing the equipment, the 'expert' midwife) **Breastfeeding–it's not rocket science** ('Anyone can do it' you just need commitment, We are here if you need us, There are other 'priorities') **Breastfeeding is a relationship** (Getting to know the woman, Getting to know the baby, Prioritising women's knowledge and abilities) |
| **"The right help at the right time": Positive constructions of peer and professional support for breastfeeding.** Burns, E. and Schmied, V. (2017) | Australia | To explore the similarities and differences in breastfeeding communication styles, and language and practices used, in the first month after birth, by privately practicing midwives, working in a continuity of care model, and trained breastfeeding peer support counsellors providing support at a national breastfeeding organisation's community-based drop-in lounge. | Private midwifery group practice and a drop-in breastfeeding drop-in centre | Qualitative inquiry with discourse analysis Observed interactions | Midwives and peer support counsellors n = 9 (sample comprised of 5 privately practising midwives and 4 peer support counsellors) | 23 | **Breastfeeding is Normal** (Normalising breastfeeding challenges, Being a trained/ professional friend) **Your body, your choices** (Facilitating autonomy, Facilitating Choice, Facilitating relationship) **The right help, at the right time** |
| **Public health nurses' experiences of supporting women to breastfeed in community settings in Ireland.** Dunne, S. and Fallon, A. (2020) | Ireland | To explore Public Health Nurses' experiences of supporting women to breastfeed in community settings in Ireland. | Rural and urban community settings | Descriptive qualitative inquiry Interviews | Public Health Nurses (PHN) n = 14 | 23 | **To give them the best support** (Supporting breastfeeding is enjoyable, To support them in their decision, Letting them know we're there) **We need the help and support as well** (I have the skills to support them, The nearest breastfeeding support group, My best support is the lactation consultant) **We've only a certain amount of time** (Time is a massive factor, Big time lapse, Time bothers me big time, It's those critical times) |

(*Continued*)

**Table 1.** (*Continued*)

| Title, Author & Year | Country | Study aim/ Research question | Practice setting | Methodology & Method | Population & Sample size | COREQ score | Findings: (Themes / sub-themes) |
|---|---|---|---|---|---|---|---|
| **A critical ethnographic study of encounters between midwives and breast-feeding women in postnatal wards in England. Dykes, F. (2005)** | England | To critically explore the nature of interactions between midwives and breast-feeding women within postnatal ward settings in northern England | Post-natal wards | Critical ethnography Interviews | Midwives n = 39 | 20 | **Taking time and touching base** (Communicating temporal pressure, Routines and procedures, Disconnected encounters, Managing breastfeeding women, Rationing information) |
| **Breastfeeding Initiation: An in-depth qualitative analysis of the perspectives of women and midwives using Social Cognitive Theory. Edwards *et al.* (2018)** | Scotland | To explore women's and midwives' expectations, knowledge and experiences of breastfeeding initiation, including skin-to-skin contact and instinctive behaviour. (Definition of initiation is breastfeeding within the first 48 hours) | Baby Friendly Initiative accredited community, labour ward and post-natal wards | Qualitative inquiry using Social Cognitive theory for analysis Interviews | Midwives n = 18 | 23 | **Expectations Knowledge Experiences** |
| **Midwives in the UK: An Exploratory Study of Providing Newborn Feeding Support for Postpartum Mothers in the Hospital Furber, C.M. and Thomson, A.M. (2007)** | England | To explore English midwives' views and experiences of providing newborn feeding support. | Two non-BFI accredited maternity hospitals | Qualitative inquiry using Grounded Theory principles Interviews | Midwives n = 30 | 25 | **Surviving Baby Feeding** (Demands on Time, Coping with newborn feeding in the hospital) |
| **The Emotions of integrating breastfeeding knowledge into practice for English Midwives: A qualitative study Furber, C.M. and Thomson, A.M. (2008a)** | England | To discover the views of English midwives in relation to their breastfeeding support role | Two non-BFI accredited maternity hospitals | Qualitative inquiry using Grounded Theory principles Interviews | Midwives n = 30 as per Furber and Thomson (2007) | 21 | **Surviving baby feeding: Emotionalisation of breastfeeding** (Anger in practice, Fear in practice, Sadness in practice, Happiness in practice) |
| **Breastfeeding practice in the UK, Midwives' perspectives Furber, C.M. and Thomson, A.M. (2008b)** | England | To discover the views of English midwives in relation to their breastfeeding support role | Two non-BFI accredited maternity hospitals | Qualitative inquiry using Grounded Theory principles Interviews | Midwives n = 30 as per Furber and Thomson (2007) | 24 | **Surviving baby feeding: Directing Feeding** Doing well with Feeding (Communicating sensitively, Facilitating breastfeeding, Reducing conflicting advice) |
| **Factors influencing the sustainability of volunteer peer support for breastfeeding mothers within a hospital environment: An exploratory qualitative study Hopper, H. and Skirton, H. (2016)** | England | To establish the sustainability of a volunteer peer support service for new breast-feeding mothers within a hospital environment from the perspective of the volunteers and ward staff. | Hospital maternity ward | Descriptive qualitative inquiry Interviews | Maternity ward staff n = 16 (sample comprised of 6 Peer supporters and 10 ward staff who were trained to support women to breastfeed: midwives, maternity care assistants and nursery nurses working on maternity ward, role prevalence not provided). | 21 | **What peer supporters brought to the maternity ward What motivated the peer supporters Factors contributing to the sustainability of the service.** |

(*Continued*)

**Table 1.** (Continued)

| Title, Author & Year | Country | Study aim/ Research question | Practice setting | Methodology & Method | Population & Sample size | COREQ score | Findings: (Themes / sub-themes) |
|---|---|---|---|---|---|---|---|
| **Midwives' experiences of helping women struggling to breastfeed.** Lawton, K. (2016) | England | To explore midwives' experiences of helping women who were struggling to breastfeed. | Post-natal wards and community settings | Descriptive phenomenology Interviews | Midwives n = 5 | 16 | **Time poverty** (Breastfeeding support and time hierarchy, conditional vs unconditional motivation) **The impact on midwives on being with women** (Emotional impact on midwives, Giving permission to women to make the 'wrong decision', Empathising with women, How midwives are seen by women in their care) **Professional Integrity** (The questioning of personal professional credibility, Confidence in personal practice, Accountability) |
| **Care during Breastfeeding: Perceptions of Mothers and Health Professionals** Lucchini-Raies *et al.* (2019) | Chile | To know the perceptions of mothers and health professionals in relation to the care provided and received during breastfeeding at primary health care level. | Primary care level family health centres. | Qualitative Inquiry Interviews | Primary Healthcare professionals n = 20 (sample comprised of nurses, midwives and family physicians– data on role prevalence not provided). | 20 | **Influence of previous care and support experiences during the breastfeeding process Importance of the context within which care is framed Addressing emotions to establish trust between professionals and mothers** |
| **Using evidence in practice: What do health professionals really do? A study of care and support for breastfeeding women in primary care** Marshall *et al.* (2006) | England | To examine the use of knowledge, and in particular, of evidence derived from research, in interactions between midwives or health visitors and breastfeeding women | Community settings in rural, suburban and urban settings | Ethnography Observations and interviews | Midwives and Health visitors n = 18 (sample comprised of 9 community midwives, 9 health visitors) | 17 | **Professional experience Formal courses and education Knowledge from research and Evidence Based Practice Knowledge from policies and guidelines Using colleagues as a source of knowledge Personal experience Knowledge use in practice** |
| **Maternal-newborn nurses' experiences of inconsistent professional breastfeeding support** Nelson (2007) | U.S.A. | To describe the meaning and significance of the common, essential elements of inconsistent professional breastfeeding support as revealed through the experiences of maternal-newborn nurses in a hospital setting | nursery post-partum, labour and delivery settings | Existential (descriptive) phenomenology Interviews | Maternal newborn nurses n = 12 | 21 | **Inconsistencies still exist but things are changing A need for "buy in" There is no escaping personal experience What works for one does not work for all Time impacts on recommendations A privileged vantage point My job" what it is and what it is not " After all, breastfeeding is maternal choice** |

*(Continued)*

**Table 1.** (Continued)

| Title, Author & Year | Country | Study aim/ Research question | Practice setting | Methodology & Method | Population & Sample size | COREQ score | Findings: (Themes / sub-themes) |
|---|---|---|---|---|---|---|---|
| **Exploring how IBCLCs manage ethical dilemmas: a qualitative study Noel-Weiss *et al.* (2012)** | U.S.A. and Canada | To explore ethical dilemmas experienced by IBCLCs, especially, how they manage such dilemmas. | Hospitals, private practice, public health and community settings | Qualitative inquiry Interviews | IBCLCs n = 7 | 23 | **Staying mother-centred** (Recognising an ethical dilemma or issue, Identifying context, Determining the IBCLC choices, Strategies used, Results and choices the mother made, Follow-up and actions to avoid similar situation) |
| **The supporting role of the midwife during the first 14 days of breastfeeding: A descriptive qualitative study in maternity wards and primary healthcare. Swerts *et al.* (2019)** | Belgium | To gain an in-depth understanding of the supporting role of midwives in breastfeeding support during the first two weeks after birth and compare it to the needs of the breastfeeding women. | 2 Hospital settings (one with BFI accreditation and one without) and home settings in public and private midwifery care. | Qualitative inquiry Observation and focus groups | Midwives n = 21 (observations, of which n = 10 participated in subsequent focus group) | 23 | Focus of the midwife Approach and expectations Hands on–hands off Observing Informing Coaching and supporting Relationship between midwife and mother |
| **Barriers to breastfeeding: a qualitative study of the views of health professionals and lay counsellors Tennant *et al.* (2006)** | England | To ascertain the views of staff about their breastfeeding support skills, and their own attitudes to supporting breastfeeding mothers, to inform the development of a training intervention that would address these attitudes as well as build knowledge. | General Practice, Health clinics and National Childbirth Trust (NCT) lay counsellor settings | Qualitative Inquiry Interviews | Health visitors, midwives and NCT breastfeeding counsellors n = 12 (sample comprised of 7 health visitors, 3 midwives and 2 NCT breastfeeding counsellors) | 20 | **Helping women make a decision about feeding Factors shaping professional practice Access to training** |
| **Medicalizing to demedicalize: Lactation consultants and the (de)medicalization of breastfeeding Torres, J.M. (2014)** | U.S.A. | Research questions: 1) To what extent, and in what ways, do lactation consultants work towards demedicalization? 2) How do lactation consultants balance demedicalization with their role as the clinical managers of breastfeeding? | Hospital inpatient and outpatient settings | Ethnography Observation and interviews | IBCLCs and doctors, midwives and nurses n = 28 (sample comprised of 19 IBCLCs and 9 healthcare professionals who were doctors and nurses–data on role prevalence not provided). | 16 | **Demedicalizing breastfeeding** (pathology, technology and medical control) **Medicalizing breastfeeding Medicalizing to demedicalize** |
| **Personal Infant Feeding Experiences of Postpartum Nurses Affect How They Provide Breastfeeding Support. Wright, A.I. and Hurst, H.M. (2018)** | U.S.A. | To describe the experiences of postpartum nurses when feeding their own infants and explore how these experiences influence the breastfeeding support they provide to new mothers. | Hospital postpartum unit | Qualitative inquiry Interviews | Postpartum nurses n = 9 | 25 | **Personal infant feeding experiences** (The decision to breastfeed was clear, Early breastfeeding was challenging, Breastfeeding after the maternity hospitalization was characterised by love or struggle, Coming to terms with the breastfeeding experience was easy for some and difficult for others) |

**Table 2. Theme development.**

| Examples of free codes that contribute to descriptive themes | Descriptive themes *with associated CERQual assessment of confidence* | Analytical themes *and summary statements* |
|---|---|---|
| Personal experience<br>Beliefs about breastfeeding<br>Prioritising breastfeeding<br>Sources of information<br>Feedback | Personal breastfeeding experience *(high confidence)*<br>Belief in the value and process of breastfeeding *(high confidence)*<br>Knowledge for practice *(moderate confidence)* | **A personal philosophy of breastfeeding support**<br>*The experience of having a personal philosophy of breastfeeding support promotes individual practice styles* |
| Referral<br>Actions of others<br>Undermining others<br>Inconsistent advice | Collaboration *(moderate confidence)*<br>Inconsistency in care and advice *(moderate confidence)*<br>Opinions of others *(low confidence)* | **Teamwork and tensions in practice**<br>*The experience of disconnection from or collaboration with colleagues undermines or progresses breastfeeding support practice* |
| Time constraints<br>Resource to enact role<br>Priority of breastfeeding support | Time and resources *(high confidence)*<br>Organisational values *(moderate confidence)*<br>Expectation of role *(low confidence)* | **Negotiating organisational constraints**<br>*The experience of negotiating organisational expectations and constraints impacts role-enactment* |
| Mother's confidence<br>Mother's commitment<br>Influences on mother<br>Communication with mothers<br>Trusting relationship | Perceptions of mother's breastfeeding reality *(low confidence)*<br>Relationship and communication *(moderate confidence)* | **Encounters with breastfeeding women**<br>*The experience of interacting with breastfeeding women fosters a tailoring of support to women's perceived needs* |

45, 58, 60–63] which also had high confidence as a finding, and *Knowledge for practice* [39, 40, 42, 44, 45, 47, 48, 56, 57, 59–64] in which there was moderate confidence in the finding.

**Personal breastfeeding experience.** Participants frequently described having a personal understanding of 'what matters' and 'how-to' in breastfeeding support, borne of prior experience. Both positive and negative personal breastfeeding experiences increased empathy for breastfeeding women. Participants tried to help women avoid the physical discomfort or psychological distress that they had encountered personally, either encouraging women that breastfeeding difficulties were surmountable, or not. Participants spoke of giving women permission to stop breastfeeding or supplement with formula, or spending time and offering breastfeeding tips and tricks learned from personal experience: *"Before I had [my baby] it was like, "Oh, you want formula? Okay, I'll go get it." It was more patient satisfaction versus "what was your intention when you came in? You wanted to breastfeed. I will help you become successful in breastfeeding" [60]* (Post-partum nurse in a hospital setting). The above quote illustrates how personal experiences of breastfeeding could both inform and transform practice.

**Belief in the value and process of breastfeeding.** There was little diversity across the studies in terms of belief in the value and process of breastfeeding. Most participants spoke of the importance and value of breastmilk and/or breastfeeding to a mother/baby dyad: *". . .midwives focused on ensuring that the infant had sufficient access to 'liquid gold'. Midwives drew on their 'expert' knowledge to introduce a range of techniques and technology to ensure that the infant received breastmilk"* [61]. Although belief in the value of breastfeeding was widely represented in the data, beliefs about the skills involved in breastfeeding varied. Some participants believed that breastfeeding was a natural skill that anyone could do if they were prepared to work at it, whereas others saw breastfeeding as a technical skill that women had to be taught.

**Knowledge for practice.** Participants spoke of the personal and vicarious learning experiences that informed their practice. Experiential knowledge, attending training courses, following guidelines and policies, engagement with research and learning from colleagues were all identified as influential. Participants selected what information, gained from experience, they felt may be useful in different situations. This was then tested against knowledge from prior breastfeeding support episodes and input from colleagues. However, working in accordance

with guidelines resulted in some participants expecting to physically intervene in breastfeeding support: '. . .*midwives described an expectation of giving "hands on" help with attachment in the labour ward. . . This may arise in part from BFI guidelines [which recommend skin-to-skin and breastfeeding within the first hour of birth]*' [45]. The experience of working to guidelines influenced the physical approach to support.

## Teamwork and tensions in practice

Data from 14 papers contributed to this analytical theme, comprised of 3 descriptive themes: *Collaboration* [48, 54, 55, 64] in which there was moderate confidence in the finding, *Inconsistency in support and advice* [45, 48, 54, 55, 58, 64] in which there was also moderate confidence and *Opinions of others* [44, 45, 47, 55, 56, 61] in which the authors had low confidence.

**Collaboration.**   Positive collaborative experiences included effective communication with others involved in breastfeeding support, a visible and effective referral system in place and respectful relationships between providers. Good teamwork with sharing of knowledge facilitated breastfeeding support provision. Nevertheless, participants often reported problems with collaboration and referral, either with the availability of staff such as lactation consultants only working certain shifts, or with other support providers waiting too long before referring a mother to expert help. The following quote highlights the frustration felt by a lactation consultant when late referrals impeded the ability to provide effective support to women experiencing breastfeeding difficulties: *"they're usually train wrecks"* [54]. This reveals a sense of despair that other breastfeeding providers do not refer women in a timely manner so that early problems may be resolved more easily.

**Inconsistency in support and advice.**   Inconsistency in support and advice was a common theme. Participants undertake damage limitation techniques to reassure the woman that what they perceive as inconsistent advice is a normal aspect of the breastfeeding learning journey. Managing conflicting and potentially damaging input from others resulted in trying to achieve a balance between offering effective support and not wishing to undermine a mother's belief in other breastfeeding supporters and healthcare providers. This was an uncomfortable experience for some participants.

**Opinions of others.**   Differing philosophies of breastfeeding support between providers impeded effective provision. Participants experienced intimidation and disapproval from some colleagues if they devoted time to breastfeeding support. Peer opinion also facilitated a more physical direct approach: '. . .*being able to competently attach an infant, often referred to as 'having the knack', was a highly prized and sought-after skill that afforded some midwives a sense of status within their professional peer group*' [61]. Experiences of perceived credibility with colleagues therefore influenced how breastfeeding support was provided.

## Negotiating organisational constraints

Participants in 11 papers contributed to the 3 descriptive themes informing this overall analytical theme: *Time and resources* [40, 42, 44, 45, 47, 48, 54–56, 58, 61, 62] which had high confidence in the findings, *Organisational values* [45, 47, 54, 55] in which there was moderate confidence in the findings and *Expectation of role* [41, 45, 47, 56, 58] in which the authors had low confidence as a finding when assessed using CERQual.

**Time and resources.**   The impact of being under-resourced in terms of time pressures and staff shortages prevented participants from providing optimal breastfeeding support. Participants spoke about stepping in to physically attach babies to the breast in the hope that it would save time and free them to attend to other clinical duties. In acute settings it was common for staff to default to such time-saving approaches to breastfeeding support even when there was no pressure on resources. Conversely, some participants who independently organise and

control their workload, for example privately practising midwives, were able to spend time to optimise the support given to breastfeeding women.

**Organisational values.** The low priority given to breastfeeding as a maternity care issue was identified as something which hinders good breastfeeding support practices. One midwife in a hospital setting spoke of decision-making in terms of breastfeeding support and available resources: *"Sometimes you have to say I'll send a midwife out, but you know that's a resource that is precious. All you can do is make sure they've got a visit the next day if it's the middle of the night. [Pause] But that's a long gap and that's not her answer at that time"* [47] (midwife providing telephone support on night shift). When the organisation is under-resourced breastfeeding support is not always viewed as a worthy recipient of an organisation's resources.

**Expectation of role.** This descriptive theme relates to the complex issue of being a breastfeeding supporter who is on one hand tasked with promoting and protecting breastfeeding, and on the other, being respectful of women's choices. One maternal newborn nurse in a postpartum unit reported *"Breastfeeding shouldn't be a hard sell. . .I mean my job is not to push somebody"* [58] demonstrating that self-perception of role led to decisions about how much encouragement to give to women.

## Encounters with breastfeeding women

Participants in all but one of the papers contributed to this analytical theme relating to interactions with breastfeeding women, comprised of 2 descriptive themes: *Perceptions of women's breastfeeding reality* [42, 44, 45, 47, 48, 54–56, 58, 61–64] in which there was low confidence in the finding, and *Relationship and communication* [39, 41, 42, 44, 47, 48, 54, 56–58, 60–65] in which there was moderate confidence in the finding. Encounters with women were opportunities to assess the woman's individual needs and also her commitment to breastfed. This information then influenced how support was provided in terms of motivation to provide support and time spent with women. However, capturing the woman's wishes required open communication and some connection or relationship to be established.

**Perceptions of mothers breastfeeding reality.** Perceptions of the mother's motivation and capability in relation to achieving breastfeeding goals were early signs used by the provider to gauge how to provide support. Some participants spoke of the impact of a mother's commitment on their own motivation in practice, for example midwives in a hospital setting: *"the women have to have some sort of commitment as well. If they don't have that commitment, then there's no point in us busting our gut to do it, either"* [61]. Thus some participants made value judgements on breastfeeding women to assess whether it was a worthwhile priority for care. This was not reported in settings were women actively sought support, for example in drop-in clinics or during consultation with a lactation consultant.

**Relationships and communication.** Some participants spoke about their sensitivity to women's views on their practice which influenced how information was conveyed: *'Sometimes you know you've to be careful that it doesn't come across very dictatorship'* [55]. They were aware of the potential vulnerability of women and the need to communicate sensitively. Relaxed and companionable relationships were used to dismantle a novice/expert dynamic in breastfeeding support in some community settings and private practice. Encounters that respected breastfeeding women's autonomy facilitated breastfeeding support that did not involve touching the woman or "doing for" her.

## Thematic synthesis

The thematic synthesis resulting from the interpretation of the four analytical themes demonstrates that trained breastfeeding support providers carry a philosophy of breastfeeding

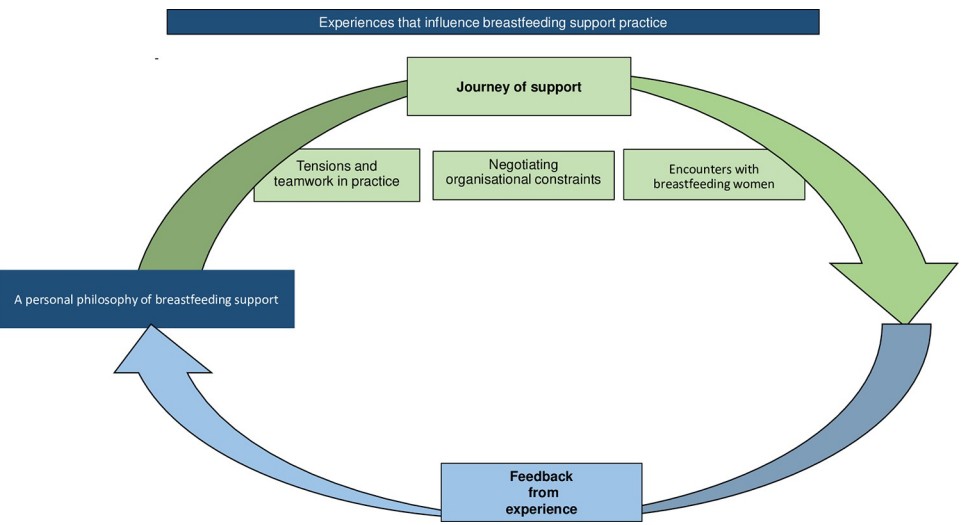

**Fig 2. Thematic synthesis.**

support with them as they start out on a journey to provide support. Findings indicate that providers have an established philosophy of support informed by personal and vicarious breastfeeding experiences, a belief (or not) in the value of breastfeeding, and preferences for "what works" in breastfeeding support practice. The philosophy of breastfeeding support develops over time with accumulation of experience. Training may shape an initial philosophy derived from participants' personal and socio-cultural reference points, but it is further developed by the experiential knowledge from practice and any new personal and vicarious breastfeeding support experiences. Reflection on and feedback from experience therefore inform the philosophy of support in a feedback system, either re-enforcing existing beliefs or introducing new information for future practice. The philosophy of support can be suppressed or enabled depending on the immediate context in which the support is being provided. There is fluidity in how the experiences affect eventual support provision, as supporting individual women in specific settings is dynamic and responsive to the overall context within which care is given. The Thematic Synthesis is represented in Fig 2.

## Discussion

Three research questions were posed in this review; what is known about experiences that influence breastfeeding support practices, how those experiences are influential and how experiences impede or facilitate support provision. This section of the paper will initially explore how the research questions have been addressed and continue with a discussion of results in the context of the wider evidence base. In our view this is the first qualitative evidence synthesis of research reporting on breastfeeding and breastfeeding support related experiences that influence the practice of a range of trained breastfeeding support providers. From the inclusion of 21 papers in the systematic review four analytical themes were generated which illustrate experiences that influence how breastfeeding support is valued, prioritised, and delivered in practice: *A personal philosophy of breastfeeding support*, *Teamwork and tensions in practice*, *Negotiating organisational constraints*, and *Encounters with breastfeeding women*. Findings reveal that exposure to and engagement with these diverse experiences cut across both professional and peer role-types, and apply across maternity and family care settings. Regardless of training undertaken, the life history of the provider and the context bound nature of

breastfeeding support practice influence how providers enact their role. A philosophy of breastfeeding support develops over time. Positive and resilient philosophies of breastfeeding support are informed by positive personal experiences, belief in the value and process of breastfeeding, service models that enable open communication with women, adequate resourcing in terms of workforce and time to spend with women, and appropriate information sources to inform practice.

The second research question focused on how experiences influence practice. Findings indicate that a range of experiences influence provider motivation and the approach to support. Decisions about whether to step away to leave a woman to learn about breastfeeding herself, whether to stay with a woman and spend time discussing her breastfeeding, or whether to take over and intervene are underpinned by the provider's own philosophy of support, their experience of colleagues breastfeeding knowledge, skills and attitudes, the resourcing of services and experiences of prior interactions with breastfeeding women. These experiences ebb and flow in their contribution to practice which can both hinder and facilitate breastfeeding support provision. Supporting a particular mother, with particular colleagues, in a particular organisation influences behavioural aspects of how support is provided.

Finally, the review sought to explore which experiences facilitate or impede support provision. Having a belief in the value of both breastmilk and the breastfeeding relationship, having positive personal experiences including having overcome difficulties, and use of a wide range of knowledge sources (including women's feedback about support received) appear to facilitate breastfeeding support provision. Honest, trusting relationships with women, respectful collegiate relations, peer learning and providing support in well-resourced organisations which visibly value breastfeeding also facilitate compassionate evidence-based breastfeeding support. Negative breastfeeding experiences or no breastfeeding experience can result in doubt about the value of breastmilk and the breastfeeding relationship, particularly when mothers are struggling with feeding challenges. Vicarious experiences of colleagues' behaviours may reenforce outdated or insensitive practices. Funding cuts to breastfeeding support services and a lack of training opportunities beyond a basic level, send a signal that breastfeeding support is not valued by an organisation. Lack of personal and professional reference points about successful breastfeeding (including no feedback from women) can result from fragmented care. An implication of such lack of feedback is that there is an absence of recognition of "what works" in practice, limiting growth and progression in the philosophy of breastfeeding support.

This review extends our understanding of why breastfeeding support provision can tend toward non-standardised, inequitable delivery and why women are reporting some dissatisfaction with their breastfeeding support despite providers being trained for the role. New insights into the importance of personal and vicarious experiences of breastfeeding and breastfeeding support have emerged that require us to think differently about the influence of provider's ongoing experiences of breastfeeding and breastfeeding support, and the communities of practice in which women are supported. Findings complement prior studies reporting low confidence and learning deficits in breastfeeding support providers [66, 67]. Furthermore, findings suggest that attention be given to personal and organisational factors in addition to education when considering how best to support women to breastfeed. Prior research has reported that effective support intervention implementation is complex and culturally specific [68, 69]. This systematic review enhances our understanding of the complexity in implementing breastfeeding support interventions. The principal implication is that an individual's approach to breastfeeding support is shaped by personal preferences, which are influenced by organisational culture, wider society and the breastfeeding support education that they receive.

The key message from the first theme *A personal philosophy of breastfeeding support* is that provider approach and behaviour during breastfeeding support encounters is grounded in how they personally make sense of breastfeeding. Exposure to skilled sensitive breastfeeding support in practice, evidence-based education, and opportunities to reframe negative experiences through reflection are required to develop a positive philosophy of breastfeeding support. Prior and current experiences of breastfeeding and breastfeeding support underpin what is valued in sharing information and teaching practical techniques. Various practice styles have been identified in earlier research into the woman's perspective of breastfeeding support, for example perception of the encounter by women as authentic or disconnected [16] or the provider acting as a skilled companion or a technical expert [24]. The findings of this systematic review give insight into such variations. Providers tended to either step in and intervene, distance themselves so that the woman learns about her own breastfeeding by herself, or be present with a woman to discuss her breastfeeding and provide support in a partnership approach. Time spent with women, the type of information and encouragement given, and behaviour during skilled help depend on both the provider's philosophy of support, and the ability of the provider to practice according to their philosophy within their particular workplace or volunteer setting.

The *Teamwork and tensions in practice* theme highlighted the importance of collegiate relations in practice. Development of communities of breastfeeding practice with skilled and motivated colleagues enhances teamwork and minimises tensions. Positive collaboration, efficient referral systems amongst support providers and the experience of being part of a knowledgeable team that values breastfeeding facilitates the creation of ad-hoc learning opportunities, easy access to breastfeeding expertise, and freedom to dedicate time to breastfeeding support without disapproval from colleagues. Lactation consultants were identified as a good resource for clinical staff when they were integrated into healthcare structures. These findings support the idea that informal learning and role modelling from experts in practice across multisector breastfeeding support settings are useful experiences that develop confidence in breastfeeding support skills [70]. Such opportunities should be recognised for their value in order to support the sustainability of this model [71]. The challenge now is to deliver effective breastfeeding services which foster peer learning in practice from valued experts, ease of referral to specialist breastfeeding support and everyday exposure to positive breastfeeding support provision from colleagues.

Attention must also focus on the influence of environments with tensions in practice due to differences in individual philosophies of breastfeeding support. The findings demonstrate that support providers can feel pressure from colleagues to underplay their breastfeeding support role or deviate from the evidence-base to free up time for other duties. This adds to existing evidence that experiencing colleagues outdated practice [72] and disjointed services [73] impede effective breastfeeding support provision. There is a need for ongoing breastfeeding education in practice with opportunity for providers to reflect on and in practice together. Pragmatically, multi-sector case study presentations could be scheduled as part of continuing breastfeeding education, with collaborative learning and exposure to positive examples of how skilled professional and peer support can enable women to overcome breastfeeding challenges. The importance of multisector collaborative working supports a recent qualitative evidence review into experiences of breastfeeding peer support [74] which proposed that tension between peer supporters and health professionals can be overcome through building trusted relationships in integrated services. This systematic review extends knowledge about the benefits of multisector working as experiencing the enthusiasm and skill of a variety of motivated providers allows others to develop their own confidence and skill in practice, benefitting women and ensuring that integrity in breastfeeding support services is upheld.

The third theme *Negotiating organisational constraints* illustrates how underfunding and the low prioritisation of breastfeeding support services limits role-enactment. Full implementation and funding of evidence-based breastfeeding support interventions is imperative for providers to experience services that value breastfeeding. Consistent with the literature, participants from studies included in this review reported that working under organisational constraints, especially time pressures, impedes practice [75–77]. The experience of providing support in well-funded services with adequate providers available enables provision of meaningful breastfeeding support because time, training and informal learning opportunities are available. Findings are consistent with components of the Breastfeeding Gear Model [78]. This tool is a multi-level and multi-sector approach to scaling up intervention implementation and identifies resourcing and political will as indispensable factors that enable sustainable and effective evidence based breastfeeding support intervention implementation [69]. If providers experience under-resourcing they make decisions about care priorities and breastfeeding support is not prioritised. These findings support the work of other studies linking understaffing and lack of time with lack of breastfeeding support provision [75, 79, 80] and women's perceptions that providers are unable to provide breastfeeding support due to a lack of time and resources within healthcare organisations [75, 77]. Funding cuts to breastfeeding support services are therefore not only detrimental to women's breastfeeding experiences [81] but also detrimental to provider development. Such cuts to services impede exposure to breastfeeding that could influence a provider's philosophy of support. In addition, when organisations do not prioritise breastfeeding, the lack of breastfeeding education beyond a basic level limits new knowledge development, sending a signal that breastfeeding support is not a priority and providers lose motivation to enact their role.

The final theme *Encounters with breastfeeding women* describes how the experience of interacting with breastfeeding women enables support to be tailored to women's perceived needs. Models of relational continuity in breastfeeding support should be promoted as these foster individualised breastfeeding support provision. It is known that making a connection with women, and communicating openly about breastfeeding goals, enables effective breastfeeding support provision [82, 83]. Overall this systematic review strengthens the idea that support which involves insight into the woman's needs and preferences about breastfeeding can be facilitated by relational continuity of carer [83, 84]. Relationship has previously been highlighted in the literature as an important component of breastfeeding support [76, 85]. These findings also concur with research demonstrating the complexities in providing individualised breastfeeding support [86] because providers want to support breastfeeding yet avoid being viewed as 'bullies' by women [87]. A review of models of care provision could inform which models prioritise open communication of women's goals, and informed and sensitive assessment of the need for individualised care during each woman's breastfeeding experience. These findings support the evidence that women display greater autonomy about breastfeeding issues when the relationship with the provider enables exploration of options together [88].

This thematic synthesis of published qualitative research has highlighted that experiences in practice and in personal lives can both facilitate and impede breastfeeding support provision. Providers need adequate support to enact their role, and to understand how their positioning shapes their values and contributes to their individual philosophy of breastfeeding support. Context specific experiences may be modifiable through improved prioritisation and resourcing of breastfeeding support services, development of breastfeeding communities of practice within organisations enabling multidisciplinary and multisector learning opportunities, and appropriately implemented and funded relational models of care. Prior personal experiences and challenges in practice that negatively influence support may be reframed through sensitive

reflection during formal breastfeeding education and informal debriefing with other support providers in a well-resourced and educated team. Breastfeeding support interventions should be based on evidence of outcome effectiveness [3] women's perspectives [89, 90] and as this review demonstrates, the context within which the trained provider experiences breastfeeding and breastfeeding support. Continued efforts are needed to enhance experiences which enable effective support, such as learning from experts in practice, being part of an organisation that values breastfeeding, and time to spend with women.

## Implications for practice

The overall contribution of this systematic review of qualitative evidence is a deeper understanding of experiences which influence how breastfeeding support is provided, and the significance of such experiences on the approach taken within support encounters. There are three specific implications for practice. Firstly, development of a multisector breastfeeding team in maternity and family services can harness the enthusiasm and expertise of self-selected healthcare professionals, trained peer supporters and lactation consultants. Shared learning through the integration of regular presentations of case histories to the wider multidisciplinary team will enhance potential for continued development and professional support networks. These teams, working alongside other healthcare professionals should ensure ready access to expertise and referral, and enable other providers to witness confident timely breastfeeding support provision. Secondly, developing and appropriately implementing models of service provision that prioritise relational continuity can enable exposure to longer term routine breastfeeding journeys. Experiencing knowledge of the outcomes of one's support can build confidence in practice, influencing belief in the value and process of breastfeeding. This will provide the opportunity to build wisdom in the approach to take with certain women, for example when to step in and intervene and when to encourage women to manage their breastfeeding independently underpinned with provider support. Thirdly, consistent full implementation and resourcing of evidence-based breastfeeding support services specifically, and maternity and family services in general, can help organisations ensure that providers are able to enact their role.

## Future research

This systematic review of qualitative evidence provides evidence that the context in which women are supported with breastfeeding in terms of working with colleagues, resourcing and valuing of breastfeeding services, and time in partnership with women has an important and ongoing influence on the development of breastfeeding support knowledge and practice. Due to heterogeneity amongst the included papers, further research with more focus on practice context and the factors which influence practice from the perspective of a range of providers in accredited settings is warranted, for example UNICEF Baby Friendly accreditation which is an international standard of best practice in multi-level breastfeeding support [43]. The potential to optimise the positive factors and minimise the negative factors influencing practice in specific settings may be addressed through evidence-based provider education and service design.

## Strengths & limitations

This review has demonstrated consistency across included papers of common experiences that influence the practice of breastfeeding support providers. Quality appraisal of the studies, data analysis, theme development and application of the CERQual tool have been described in detail in the text and supplementary material, increasing transparency for the reader and trustworthiness of the review. There was high confidence in the descriptive themes *Personal*

*breastfeeding experience*, *Belief in the value and process of breastfeeding* and *Time and resources* due to the adequacy of relevant coherent data and only minor concerns about possible methodological bias.

Methodological limitations within studies include lack of transparency in reporting. No study fulfilled all of the COREQ reporting criteria, with researcher reflexivity under-reported in most studies. Limitations of the review findings in their entirety were assessed using the CERQual tool. Concern about inadequate, irrelevant or incoherent data resulted in the downgrading of some of this review's descriptive themes from high confidence to moderate or low confidence in findings. There is a paucity of general practitioner, health visitor and trained peer supporter representation in the studies, midwives and lactation consultants predominate in the sample. Some participants identifying as lactation consultants held dual roles, for example as midwife or paediatrician. For the purpose of this study participants were identified as the role-type of the study inclusion criteria in which they participated. Sub-group analysis by role-type was not possible due to heterogeneity of the population. Heterogeneity of methodologies, population demographics and the settings of breastfeeding support provision represented in this review has limited the synthesis and as such the wider transferability of findings. Despite conducting an extensive database search it is possible that not all relevant records were retrieved due to the use of search limiters such as English language and publication date.

## Conclusion

This systematic review has identified personal, professional and workplace experiences which inform the development of a philosophy of breastfeeding support, and experiences which facilitate or hinder expression of that philosophy. This evidence contributes to our understanding of why breastfeeding support interventions can be experienced in a variety of ways by women. The findings suggest the more exposure to effective breastfeeding support that a provider experiences, the richer their philosophy of support becomes. This qualitative evidence synthesis adds to the growing body of literature indicating that successful implementation of complex breastfeeding support interventions requires a deeper understanding of the relational aspects of support between the provider and the woman, and the context within which support is provided.

## Supporting information

**S1 Checklist. PRISMA 2020 checklist.**
(DOCX)

**S1 Table. Table of search histories.**
(DOCX)

**S2 Table. Prevalence of themes.**
(DOCX)

**S3 Table. CERQual evidence table.**
(DOCX)

**S1 File. COREQ appraisal of reporting.**
(DOCX)

## Author Contributions

**Conceptualization:** Mary Jo Chesnel.

**Supervision:** Maria Healy, Jenny McNeill.

**Writing – original draft:** Mary Jo Chesnel.

**Writing – review & editing:** Mary Jo Chesnel, Maria Healy, Jenny McNeill.

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
