## [Decision Letter · Decision Letter 0]

15 Mar 2022

PONE-D-21-32296Supporting breastfeeding women: a thematic synthesis of experiences which influence trained providers' practice.PLOS ONE

Dear Dr. Chesnel,

Thank you for submitting your manuscript to PLOS ONE. After careful consideration, we feel that it has merit but does not fully meet PLOS ONE’s publication criteria as it currently stands. Therefore, we invite you to submit a revised version of the manuscript that addresses the points raised during the review process.

We look forward to receiving your revised manuscript.

Kind regards,

Resham B Khatri, PhD

Academic Editor

PLOS ONE

Journal Requirements:

2. We noted in your submission details that a portion of your manuscript may have been presented or published elsewhere. Please clarify whether this publication was peer-reviewed and formally published. If this work was previously peer-reviewed and published, in the cover letter please provide the reason that this work does not constitute dual publication and should be included in the current manuscript.

4. Please include a caption for figure 1.

Reviewers' comments:

Reviewer's Responses to Questions

**Comments to the Author**

1. Is the manuscript technically sound, and do the data support the conclusions?

Reviewer #1: Partly

Reviewer #2: Yes

2. Has the statistical analysis been performed appropriately and rigorously? 

Reviewer #1: No

Reviewer #2: N/A

3. Have the authors made all data underlying the findings in their manuscript fully available?

Reviewer #1: No

Reviewer #2: Yes

4. Is the manuscript presented in an intelligible fashion and written in standard English?

Reviewer #1: Yes

Reviewer #2: Yes

5. Review Comments to the Author

Reviewer #1: This work is novel and raises relevant public health issue. Structure of the paper is well established. There is a clear aim of the study. I found that you missed PRISMA flow chart with systematic search criteria, inclusion and exclusion criteria in the figure. Please do not start sentence with number (eg line 37).

Reviewer #2: The methodology of this systematic review is well conducted and reported. It also covers an important area of breast feeding which demands in-depth understanding through qualitative research. The necessary supporting information for different steps of systematic review is provided in the supplemental file. However, discussion section could have been richer.

Following comments could help improve the quality of this manuscript.

1. Title: It is better to clearly specify that it is a systematic review of qualitative research or evidence in the title. The current title doesn’t clearly give that message to the readers

2. Abstract:

Findings- Please don’t start the sentence with number. Please also specify the total search results out of which the 21 papers were included.

3. Methods

-Please begin this section by specifying that you have followed PRISMA guidelines, and also provide details of protocol registration as mentioned in the abstract.

-Search strategy – line 123 – Please clearly specify what your PEOT is by elaborating criteria for each category.

-Though the authors have mentioned the inclusion and exclusion criteria, it is better to present it in a more structured way sooner so that the readers don’t have to search through the text to figure it out.

4. Findings- please cite the flow diagram of the findings described in the main text.

-Study characteristics- please break the paragraph.

-Pls, dedicate a paragraph to describe PRISMA flow chart and another paragraph describing the study characteristics and describe the table on characteristic of studies.

-Pls also cite the table that is included in the supplemental file.

-Technically, that table should be part of main text rather than a table in supplemental file.

-The PRISMA flow chart should also be part of the main text.

-Line 241- Subheading “Findings” here makes no sense. Please rename with what this subsection represents. Probably the themes?

5. Discussion

Discussion section needs a bit of restructuring. This is just a suggestion; the flow seems a bit interrupted which could be improved. For example, it could make sense to discuss the research questions before going to each theme. Also the paragraphs elaborating on themes seems simplistic. It doesn’t provide as much depth as it demands. While the authors have compared and contrasted the findings of each theme with other studies, it would be better if the authors could also provide deeper insights into the “why” and “how” for each theme and what the authors think can be done to improve things under each theme.

There are typing errors at some places. Please thoroughly proofread the manuscript before submitting the revised version.

6. PLOS authors have the option to publish the peer review history of their article (what does this mean?). If published, this will include your full peer review and any attached files.

Reviewer #1: **Yes: **Dr Shalik Ram Dhital, MScPH, PhD

Reviewer #2: No

---

## [Author Response · Author response to Decision Letter 0]

28 Apr 2022

Reviewer Comment 1: Please ensure that your manuscript meets PLOS ONE's style requirements, including those for file naming. The PLOS ONE style templates can be found at 

Author response: The manuscript has been edited accordingly and figures have been uploaded to PACE tool. Unfortunately we were unable to achieve permissions for the illustrations in the original graphic Figure 2 Thematic synthesis and these have been removed from the Figure. 

Reviewer comment 2: We noted in your submission details that a portion of your manuscript may have been presented or published elsewhere. Please clarify whether this publication was peer-reviewed and formally published. If this work was previously peer-reviewed and published, in the cover letter please provide the reason that this work does not constitute dual publication and should be included in the current manuscript.

Author Response: This manuscript does not represent dual publication. Only the protocol (the planned approach that would be undertaken) for this systematic review has been published [1]

Reviewer comment 3: Please note that in order to use the direct billing option the corresponding author must be affiliated with the chosen institute. Please either amend your manuscript to change the affiliation or corresponding author, or email us at plosone@plos.org with a request to remove this option.

Author response: Direct billing is not required as Queen’s University Belfast has an institutional agreement with PLOS One for publication. The corresponding author is a PhD candidate at Queen’s University and has previously published in PLOS One [1]

Reviewer comment 4: Please include a caption for figure 1. 

Author response: Edited, Line 221: Caption included 

1. Is the manuscript technically sound, and do the data support the conclusions? The manuscript must describe a technically sound piece of scientific research with data that supports the conclusions. Experiments must have been conducted rigorously, with appropriate controls, replication, and sample sizes. The conclusions must be drawn appropriately based on the data presented.

Reviewer #1: Partly

Reviewer #2: Yes 

Author comment: Data does support the conclusions as highlighted by reviewer #2 

2. Has the statistical analysis been performed appropriately and rigorously?

Reviewer #1: No

Reviewer #2: N/A 

Author comment: N/A as per Reviewer 2 

3. Have the authors made all data underlying the findings in their manuscript fully available? The PLOS Data policy requires authors to make all data underlying the findings described in their manuscript fully available without restriction, with rare exception (please refer to the Data Availability Statement in the manuscript PDF file). The data should be provided as part of the manuscript or its supporting information, or deposited to a public repository. For example, in addition to summary statistics, the data points behind means, medians and variance measures should be available. If there are restrictions on publicly sharing data—e.g. participant privacy or use of data from a third party—those must be specified.

Reviewer #1: No

Reviewer #2: Yes 

Author comment: Yes, data underlying the findings is fully available as highlighted by Reviewer #2. This manuscript is a systematic review of qualitative evidence and the published papers from which the data have been extracted are referenced. 

4. Is the manuscript presented in an intelligible fashion and written in standard English? PLOS ONE does not copyedit accepted manuscripts, so the language in submitted articles must be clear, correct, and unambiguous. Any typographical or grammatical errors should be corrected at revision, so please note any specific errors here.

Reviewer #1: Yes

Reviewer #2: Yes 

Author comment: Edited. Proof reading carefully undertaken and edits of typographical and grammatical errors amended

Reviewer #1: This work is novel and raises relevant public health issue. Structure of the paper is well established. There is a clear aim of the study. I found that you missed PRISMA flow chart with systematic search criteria, inclusion and exclusion criteria in the figure. Please do not start sentence with number (eg line 37). 

Author comment: Thank you for your comments, I have addressed the issues as follows: 

Line 219: PRISMA reference included

Line 221: Figure of PRISMA flow chart included in text (Figure 1). Inclusion and exclusion criteria now included in the PRISMA flow chart figure.

Line 163-164: The reader is directed to the supplementary file S1 Table of Search histories.

Lines 131-151: Now includes Inclusion & Exclusion Criteria

Line 41: Edited to read ‘This systematic review includes 21 papers…’

Reviewer #2: The methodology of this systematic review is well conducted and reported. It also covers an important area of breast feeding which demands in-depth understanding through qualitative research. The necessary supporting information for different steps of systematic review is provided in the supplemental file. However, discussion section could have been richer. Following comments could help improve the quality of this manuscript.

1. Title: It is better to clearly specify that it is a systematic review of qualitative research or evidence in the title. The current title doesn’t clearly give that message to the readers 

Author comment: Thank you for your comments. I have changed the title in response and Line 1 now reads ‘Experiences that influence how trained providers support women with breastfeeding: a systematic review of qualitative evidence’

Reviewer comment 2. Abstract: Findings- Please don’t start the sentence with number. Please also specify the total search results out of which the 21 papers were included. 

Author comment: Edited as requested. Line 41: The sentence no longer starts with a number, it now reads: This systematic review includes 21 papers comprising 368 participants from the 1811 records retrieved.

Reviewer comment 3. Methods. Please begin this section by specifying that you have followed PRISMA guidelines, and also provide details of protocol registration as mentioned in the abstract. 

Author response: Edited as requested Line 106-108: This sentence now reads: PRISMA guidelines were followed in conducting this review [2]. The review protocol is registered in the International Prospective Register of Systematic Reviews: PROSPERO registration number CRD42020207380 and has been published.

Reviewer comment: Search strategy – line 123 – Please clearly specify what your PEOT is by elaborating criteria for each category.

Author response: Edited and now included. In addition this information is included in the Supplementary Information file S1 Table of Search histories. 

Line 124-128: This section specifies the PEOT and now reads: A systematic search strategy was developed in collaboration with an expert subject librarian using a PEOT [27] question format: Population (trained breastfeeding support providers as per study definition above), Exposure (breastfeeding and breastfeeding support provision), Outcome (experiences that influence breastfeeding support practices), Type of study (studies that have qualitative methods or findings). 

Line 163-164 now reads: The search strategy is given in the supplementary file S1 Table of search histories.

Reviewer comment: Though the authors have mentioned the inclusion and exclusion criteria, it is better to present it in a more structured way sooner so that the readers don’t have to search through the text to figure it out.

Author response: Edited so that inclusion and exclusion criteria are presented sooner, following PEOT information. A brief summary of the inclusion and exclusion criteria has also been included in the PRISMA flow chart. Lines 131- 164: Inclusion and Exclusion criteria presented to reader sooner in the manuscript. Line 221: Figure 1 PRISMA flow chart contains a brief summary of the inclusion and exclusion criteria

Reviewer comment 4. Findings- please cite the flow diagram of the findings described in the main text. 

Author response: Edited as follows: Line 219: PRISMA referenced. Line 221: PRISMA figure cited 

Reviewer comment: -Study characteristics- please break the paragraph.

Author response: Edited. The paragraph has been split into two smaller paragraphs with the first focusing on study characteristics and the second focusing on the study quality 

Line 236-255: Study characteristics section 

Line 256-263: Critical appraisal section 

Reviewer comment: Pls, dedicate a paragraph to describe PRISMA flow chart and another paragraph describing the study characteristics and describe the table on characteristic of studies. 

Author response: Edited. Lines 214- 220: Paragraph describes the PRISMA flow chart 

Lines 236-255: Paragraph describes the study characteristics. 

Lines 228-239 describe the table of study characteristics and the first sentence now reads: Table 1 describes data extracted from each study and the study quality under the headings Title, Author, Year, Country, Study aim or research question, Practice setting, Methodology and method, Population and sample size, COREQ score and Findings (Themes/subthemes).

Lines172-179 also describe the data extraction form and study characteristics of interest. 

Reviewer comment: Pls also cite the table that is included in the supplemental file.Technically, that table should be part of main text rather than a table in supplemental file.

Reviewer response: Edited and the Study characteristics table has been moved into main text as requested rather than being a supplemental file and cited. Line 228-230: Now cites Table 1 Study characteristics in text

Line 225: Now contains Table 1 Study characteristics. 

Reviewer comment: The PRISMA flow chart should also be part of the main text.

Author comment: Edited as requested. PRISMA flow chart moved into main text as figure 1.Line 221 will now contain PRISMA flow chart in text as Figure 1

Reviewer comment: Line 241- Subheading “Findings” here makes no sense. Please rename with what this subsection represents. Probably the themes? 

Author comment: Renamed as requested, Line 256 edited to read ‘Themes’

Reviewer comment: Discussion section needs a bit of restructuring. This is just a suggestion; the flow seems a bit interrupted which could be improved. For example, it could make sense to discuss the research questions before going to each theme. 

Author response: Thank you for your comment, The discussion has been edited, restructured and additional content included as follows:Lines 422-582: The discussion has been restructured and now begins with a discussion of the research questions as requested. 

Reviewer comment: Also the paragraphs elaborating on themes seems simplistic. It doesn’t provide as much depth as it demands. While the authors have compared and contrasted the findings of each theme with other studies, it would be better if the authors could also provide deeper insights into the “why” and “how” for each theme and what the authors think can be done to improve things under each theme. 

Author response: Thank you for these suggestions. The discussion has been revised as follows: 

The discussion has been enhanced with revisions to the theme sections in lines 483-567 and inclusion of additional references [3-6] (See Response to reviewer letter for references) .

 In each paragraph discussing a theme (within this discussion section) the second sentence summarizes improvements to be made in practice in relation to each theme, and then further insights are developed for the remainder of each paragraph, for example lines 530-533. 

In addition, lines 586-602 now include three examples for practice improvement.

Reviewer comment. There are typing errors at some places. Please thoroughly proofread the manuscript before submitting the revised version. 

Author response: Edited. Supplementary information files have been re-numbered due to some information now being presented as Figures/ Tables instead of supplementary information. The PRISMA 2020 checklist was edited to reflect the page numbers in the revised manuscript. Supplementary Information now ordered as follows:

S1 Table of Search histories

S2 Prevalence of themes

S3 CERQual Evidence table

S4 COREQ Appraisal of Reporting

S5 PRISMA 2020 Checklist

---

## [Decision Letter · Decision Letter 1]

26 Jul 2022

PONE-D-21-32296R1Experiences that influence how trained providers support women with breastfeeding: a systematic review of qualitative evidencePLOS ONE

Dear Dr. Chesnel,

Thank you for submitting your manuscript to PLOS ONE. After careful consideration, we feel that it has merit but does not fully meet PLOS ONE’s publication criteria as it currently stands. Therefore, we invite you to submit a revised version of the manuscript that addresses the points raised during the review process.

We look forward to receiving your revised manuscript.

Kind regards,

Resham B Khatri, PhD

Guest Editor

PLOS ONE

Journal Requirements:

Reviewers' comments:

Reviewer's Responses to Questions

**Comments to the Author**

1. If the authors have adequately addressed your comments raised in a previous round of review and you feel that this manuscript is now acceptable for publication, you may indicate that here to bypass the “Comments to the Author” section, enter your conflict of interest statement in the “Confidential to Editor” section, and submit your "Accept" recommendation.

Reviewer #1: All comments have been addressed

Reviewer #2: All comments have been addressed

2. Is the manuscript technically sound, and do the data support the conclusions?

Reviewer #1: Yes

Reviewer #2: Yes

3. Has the statistical analysis been performed appropriately and rigorously? 

Reviewer #1: Yes

Reviewer #2: N/A

4. Have the authors made all data underlying the findings in their manuscript fully available?

Reviewer #1: Yes

Reviewer #2: Yes

5. Is the manuscript presented in an intelligible fashion and written in standard English?

Reviewer #1: Yes

Reviewer #2: Yes

6. Review Comments to the Author

Reviewer #1: Dear Author

This is a novel work that you have addressed reviewers' comments nicely. However, Please final check the language before the publication of this manuscript.

Reviewer #2: Overall, the authors have addressed the comments and the manuscript looks better. However, there are still a few points that could help further in improving the quality. Some comments are as follows. The line numbers mentioned below refers to numbers in the file with track changes.

Abstract-

Line 45-46 findings –is it 1811 record screened or retrieved? Perhaps screened would be the more appropriate term.

It is also better to report the impression of quality assessment/ risk of bias assessment of the included studies.

Pls refer to PRISMA 2020 checklists for abstract

https://prisma-statement.org//documents/PRISMA_2020_abstract_checklist.pdf

to ensure all the information needed to be reported for the abstract are included

Methods

For systematic review of qualitative studies, we focus on PICo which stands for population, interest, and context as the outcomes are not directly measurable in qualitative studies unlike for quantitative studies. The term “Outcome” could be replaced with Context instead. E.g search strategy Lines 136-138 and Data Extraction Line 215 of the tack changes file.

Findings

Included studies – as per the PRISMA checklist, it is also important to mention the reasons for exclusion for the studies excluded after full-text screening.

The tables should follow the description (text) and not precede it. Pls re-order the description of Table 1 followed by the Table placed below it.

Critical appraisal

Line 292- Please provide the full name of the tool. CASP has different tools for different study designs. So please specify it as the CASP Qualitative Studies Checklist if that was the tool used.

The critical appraisal in the results section should elaborate on the overall impression based on CERQual Evidence Profile and COREQ. What were the areas that were not reported by most studies? How many studies showed low and moderate confidence and what they lacked in their methods which lead the studies to be rated as moderate or low? Also what were the strengths of most studies in terms of critical appraisal?

Limitations

Limitations need to be elaborated. As per the PRISMA the limitations can be divided into two parts. The first part should focus on the limitations within the included studies such as their methodological biases, overall impression from critical appraisal. What the studies lacked which may have affected the interpretation in this systematic review? There seemed to be a few studies that showed low to moderate confidence in CERQual Evidence Profile, which need to be addressed as limitation of the included studies. Also not all studies fulfilled all the reported criteria as per COREQ.

The second part should focus on the limitations that may have arisen during the review method such as limitations in search, inclusion, language bias (as you’ve only included English studies), etc.

7. PLOS authors have the option to publish the peer review history of their article (what does this mean?). If published, this will include your full peer review and any attached files.

Reviewer #1: **Yes: **Dr Shalik Ram Dhital

Reviewer #2: No

---

## [Author Response · Author response to Decision Letter 1]

17 Sep 2022

Thank you for reviewing our submission titled “Experiences that influence how trained providers support women with breastfeeding: A systematic review of qualitative evidence.” I am attaching a revised manuscript updated with the minor revisions advised and would like to thank the reviewers for their helpful suggestions which we believe has strengthened the paper. 

The request for re-submission included suggested improvements in the formatting of the document and specific comments. Detail in relation to these requirements from the reviewers is provided below.

We hope this has answered all of the comments received and appreciate the review of our paper. If further clarity is required on any aspect we are happy to address and look forward to your response on our revised version. 

This Response to Reviewers has also been uploaded in this submission in tabular format for ease of reading. Line numbers in the text below refer to the Revised track changes document.

Abstract

Reviewer comment: Line 45-46 findings –is it 1811 record screened or retrieved? Perhaps screened would be the more appropriate term 

 Author response: Edited to reflect the number of records screened by title and abstract. Of note, 1811 records were identified and duplicates were removed by Covidence software leaving 977 for screening.

Line 44-45 now reads “977 records were screened”

Reviewer comment: It is also better to report the impression of quality assessment/ risk of bias assessment of the included studies. Pls refer to PRISMA 2020 checklists for abstract https://prisma-statement.org//documents/PRISMA_2020_abstract_checklist.pdf to ensure all the information needed to be reported for the abstract are included 

 Author response: The PRISMA abstract has been reviewed and the abstract complies with all elements with the exception of the requirement for funding information which is provided in the financial disclosure statement of the PLOS submission. The abstract has been reworded slightly to ensure the PRISMA abstract checklist elements are included within the limits of the PLOS abstract word limit. 

Line 35 now reads: “including Medline and CINAHL”

Line 36-39 now read: “Studies eligible for inclusion reported professional and trained peer experiences of supporting women to breastfeed. PRISMA guidelines were followed and included studies were quality appraised using the CASP Qualitative Checklist.” 

Line 46 now reads: “Following quality appraisal, all studies were deemed suitable for inclusion”.

Methods

Reviewer comment: For systematic review of qualitative studies, we focus on PICo which stands for population, interest, and context as the outcomes are not directly measurable in qualitative studies unlike for quantitative studies. The term “Outcome” could be replaced with Context instead. E.g search strategy Lines 136-138 and Data Extraction Line 215 of the tack changes file. 

 Author response: Edited with suggestion to replace outcome with context – thank you for this suggestion. This necessitated a slight restructuring of the sentence, and replacement of the word “Outcome” with “Context” in Supplementary material S1 (Table of search histories) and Figure 1 (PRISMA flow diagram), and removal of the word “outcomes” from line 181 to ensure consistency throughout the manuscript. 

Lines 128-130 now read: “A systematic search strategy was developed in collaboration with an expert subject librarian guided by a PEOT [27] format. For the purposes of this review “context” replaces “Outcome” in the mnemonic as outcomes are not directly measurable in qualitative studies.”

Line 181 now reads: “Meaningful sections of text were extracted that identified experiences...”

Findings

Reviewer comment: Included studies – as per the PRISMA checklist, it is also important to mention the reasons for exclusion for the studies excluded after full-text screening.

 Author response: Edited to include a sentence stating reasons for exclusion as illustrated in Figure 1 PRISMA flow chart.

Line 223-226 now read: “25 were excluded for reasons of wrong population (e.g. not providing routine breastfeeding support), wrong exposure (no data on breastfeeding or breastfeeding support), wrong context (no data on breastfeeding support practices) or wrong type of study (e.g. pilot evaluations).”

Reviewer comment: The tables should follow the description (text) and not precede it. Pls re-order the description of Table 1 followed by the Table placed below it. 

 Author response: Edited so that the description precedes the table. Also text omitted in error (theme names in table) has now been included. 

Line 248-251 now read: “Data was extracted from each study comprising Title, Author, Year, Country, Study aim or research question, Practice setting, Methodology and method, Population and sample size, COREQ score and Findings (Themes/subthemes). Study characteristics and quality appraisal scores are summarized in Table 1.”

Critical appraisal

Reviewer comment: Line 292- Please provide the full name of the tool. CASP has different tools for different study designs. So please specify it as the CASP Qualitative Studies Checklist if that was the tool used.

 Author response: Edited as advised.

Line 277 now reads: “The Critical Appraisal Skills Programme (CASP) Qualitative Checklist [32] was used…”

Reviewer comment: The critical appraisal in the results section should elaborate on the overall impression based on CERQual Evidence Profile and COREQ. What were the areas that were not reported by most studies? How many studies showed low and moderate confidence and what they lacked in their methods which lead the studies to be rated as moderate or low? Also what were the strengths of most studies in terms of critical appraisal? 

 Author response: Edited to explain the overall impression of study quality. Further information has been included to demonstrate how the quality appraisal based on CASP and COREQ informed the CERQual Evidence Profile of confidence in this reviews’ findings. This explanation of confidence in our findings, informed by prior quality appraisal of the contributing studies, has been added further into the findings section as it relates to the overall research output of this review. 

Lines 281-282 now reads: “Most studies did not report on researcher reflexivity. Description of analytical methods was limited in five of the studies.”

Lines 285-286: “Overall, strengths of the studies lay in the congruence of the research aims and objectives with the study design.”

Lines 297-303: “Confidence in the review findings was assessed using the CERQual tool [34]. There was high confidence in three of the descriptive themes. Confidence was downgraded to moderate (five descriptive themes) or low (three descriptive themes) when there was concern about any of the four components of the CERQual assessment [34]: methodological limitations, coherence, adequacy of data, and relevance as reported in supplementary information S3.” 

Limitations

Reviewer comment: Limitations need to be elaborated. As per the PRISMA the limitations can be divided into two parts. The first part should focus on the limitations within the included studies such as their methodological biases, overall impression from critical appraisal. What the studies lacked which may have affected the interpretation in this systematic review? There seemed to be a few studies that showed low to moderate confidence in CERQual Evidence Profile, which need to be addressed as limitation of the included studies. Also not all studies fulfilled all the reported criteria as per COREQ. The second part should focus on the limitations that may have arisen during the review method such as limitations in search, inclusion, language bias (as you’ve only included English studies), etc. 

 Author comment: The strengths and limitations section has been edited, firstly, to highlight methodological limitations within the included studies. Secondly, the CERQual evidence profile provides a measure of confidence in the findings of the review itself once the thematic synthesis is complete, to enable transparency of the review findings for the reader. Limitations in the review are now stated in terms of confidence in findings as per CERQual assessment, the limited representations of certain roles, and limits in search including language and publication date limiters.

Lines 653-666 now read: “Quality appraisal of the studies, data analysis, theme development and application of the CERQual tool have been described in detail in the text and supplementary material, increasing transparency for the reader and trustworthiness of the review. There was high confidence in the descriptive themes Personal breastfeeding experience, Belief in the value and process of breastfeeding and Time and resources due to the adequacy of relevant coherent data and only minor concerns about possible methodological bias. Methodological limitations within the studies include lack of transparency in reporting, as no study fulfilled all of the COREQ reporting criteria, with researcher reflexivity under-reported in most studies. Limitations of the review findings in their entirety were assessed using the CERQual tool. Concern about inadequate, irrelevant or incoherent data resulted in the downgrading of some of this review’s descriptive themes from high confidence to moderate or low confidence in findings. There is a paucity of general practitioner, health visitor and trained peer supporter representation in the studies…”

Line 671-675 now read: “Heterogeneity of methodologies, population demographics and the settings of breastfeeding support provision represented in this review has limited synthesis and as such the wider transferability of findings. Despite conducting an extensive database search it is possible that not all relevant records were retrieved due to the use of search limiters such as English language and publication date.”

---

## [Editor Report · Decision Letter 2]

20 Sep 2022

Experiences that influence how trained providers support women with breastfeeding: a systematic review of qualitative evidence

PONE-D-21-32296R2

Dear Dr. Chesnel,

We’re pleased to inform you that your manuscript has been judged scientifically suitable for publication and will be formally accepted for publication once it meets all outstanding technical requirements.

Kind regards,

Resham B Khatri, PhD

Guest Editor

PLOS ONE
---

## [Editor Report · Acceptance letter]

23 Sep 2022

PONE-D-21-32296R2 

Experiences that influence how trained providers support women with breastfeeding: a systematic review of qualitative evidence. 

Dear Dr. Chesnel:

I'm pleased to inform you that your manuscript has been deemed suitable for publication in PLOS ONE. Congratulations! Your manuscript is now with our production department. 

Kind regards, 

on behalf of

Dr. Resham B Khatri 

Guest Editor

PLOS ONE